# The Role of Reference Materials in the Research and Development of Diagnostic Tools and Treatments for Haemorrhagic Fever Viruses

**DOI:** 10.3390/v11090781

**Published:** 2019-08-24

**Authors:** Giada Mattiuzzo, Emma M. Bentley, Mark Page

**Affiliations:** Division of Virology, National Institute for Biological Standards and Control, Blanche Lane, South Mimms, Hertfordshire EN6 3QG, UK

**Keywords:** haemorrhagic fever viruses, international standards, vaccines, diagnostics, serology

## Abstract

Following the Ebola outbreak in Western Africa in 2013–16, a global effort has taken place for preparedness for future outbreaks. As part of this response, the development of vaccines, treatments and diagnostic tools has been accelerated, especially towards pathogens listed as likely to cause an epidemic and for which there are no current treatments. Several of the priority pathogens identified by the World Health Organisation are haemorrhagic fever viruses. This review provides information on the role of reference materials as an enabling tool for the development and evaluation of assays, and ultimately vaccines and treatments. The types of standards available are described, along with how they can be applied for assay harmonisation through calibration as a relative potency to a common arbitrary unitage system (WHO International Unit). This assures that assay metrology is accurate and robust. We describe reference materials that have been or are being developed for haemorrhagic fever viruses and consider the issues surrounding their production, particularly that of biosafety where the viruses require specialised containment facilities. Finally, we advocate the use of reference materials at early stages, including research and development, as this helps produce reliable assays and can smooth the path to regulatory approval.

## 1. Background on Reference Materials and Their Use

The use of biological reference materials has a long history dating from the 1890s, when the potency measurement of diphtheria antitoxin was found to be inconsistent and resulted in unwanted clinical consequences [1]. The antiserum was developed by Emil von Behring at the Robert Koch Institute using serum from horses inoculated with the diphtheria bacterium and applied with notable success as passive immunotherapy in infected patients. Paul Ehrlich determined that the cause of the discrepant results was due to the variability of diphtheria toxin used in the potency measurement assays. Ehrlich resolved this by expressing the potency relative to a comparator serum preparation or standard. This standard would be used in all assays to provide an agreed and consistent quantity of the antitoxin [2]. The use of such standards in improving assay accuracy for the measurement of biological activity, which cannot be done by physico-chemical means, has been adopted ever since and advanced through the work of Sir Henry Dale in the 1920s, who developed the first international standard for insulin [3]. The quality and activity of biological medicines is, therefore, assured by this approach. Assay accuracy for the diagnosis of viral haemorrhagic fever (HF) is obviously critical so that appropriate healthcare can be provided as soon as possible. Both a false positive and a false negative result could have disastrous consequences. A false positive result might lead to an uninfected patient being quarantined with other infected individuals, and thereby run the risk of becoming themselves infected, whereas a false negative would result in the infected patient receiving no treatment at all.

Several reference materials are available and should be used according to their application in a given assay (Table 1). The World Health Organisation (WHO) International Standard (IS) is the highest order standard against which all others are calibrated. The IS should only be used for occasional (annual) calibration of the assay so that it is not exhausted too quickly. It is important to maintain the IS for a long period of time (5–10 years) to ensure a consistent and unchanging comparator, and thereby preserve traceability [4]. Secondary standards can be used more frequently for sample quantification and can be calibrated to the IS, if one is available; they also serve as assay performance monitors to assess the assay over time and to evaluate any variations introduced by different operators or tweaks to the assay protocol. Further, they facilitate assay performance comparisons between laboratories, either calibrated to the IS, *per se*, or as part of an external quality assessment scheme [5]. Internal standards are usually supplied with the assay and used in every run to monitor performance, such as sensitivity and limits of detection. It is an inevitability that an assay will “drift” over time and an internal control would not necessarily identify this problem. Therefore, external traceable standards can provide a solution. Even so, it is important to regularly (annually) calibrate the assay to the International Standard.

For quantitative assays, an arbitrary value (Unit) assigned to the calibrant is a useful way to express the results as a relative potency to the comparator; this leads to improved assay harmonisation and the ability to compare assays. This is an important concept that Ehrlich recognised, who was able to provide an elegant and effective solution to the problem of discrepant results. Reporting biological activity in physical or chemical units, such as mg or copies, therefore is neither relevant nor appropriate. The International Standard is assigned an International Unit per ampoule for this purpose [4]. In some cases, the assigned unitage of a reference preparation in unit/mL is arbitrarily chosen to be similar to the value of the physical units; the reason for this is to facilitate the end user in the transition from the metrology system in use (mg/mL, copies/mL, etc.) to the potency expressed in unit/mL

Finally, it is important to highlight what the reference standards are not intended to be used for. While they can help determine the sensitivity or cut-off of the assay, they cannot be used to determine assay specificity. Importantly, reference standards are not intended to be used as a means of assay validation—this should be conducted with a large panel of clinical samples. Furthermore, a limitation on the use of a reference material is that it will not solve substantial issues related to the assay. For instance, if the sensitivity of an assay is very low, the use of a reference material may highlight this potential problem, but it will not correct a false negative result.

## 2. Setting the Scene for Haemorrhagic Fever Virus Applications

Haemorrhagic fever virus (HFV) is a general term used to classify viruses that cause common signs and symptoms. This is largely considered to apply to the *Arenaviridae*, *Bunyaviridae*, *Filoviridae* and *Flaviviridae* families—all zoonotic RNA viruses [6,7]. While viruses within the HFV category may share a similar clinical course, their virulence, mode of transmission and incidence is more diverse. This calls for sensitive and specific diagnostic tests and a differential diagnostic protocol to be in place so that upon presentation at medical facilities patients are appropriately triaged and prescribed treatment. Traditionally, confirmatory HFV diagnosis has been performed by large reference laboratories often based outside the outbreak country, and at times on a different continent [7,8]. However, technological advances mean that it is now commonplace to train and equip laboratories in the field to run these tests or deploy mobile diagnostic laboratories within outbreak settings. The past and ongoing outbreaks of Ebola virus (EBOV) disease in Africa serve as a pertinent example. Since the outbreak in Gulu in 2000, there has been a large effort to deploy in-field and mobile diagnostics [9,10,11,12,13]. Further, during 2013–2016 EBOV epidemic in Western Africa, which the WHO declared a Public Health Emergency of International Concern (PHEIC), the U.S. Food and Drug Administration (FDA) granted emergency use authorisation for several EBOV in vitro diagnostic platforms, which were subsequently deployed in-field [7,14]. This sets a precedent for future high consequence HFV outbreaks and highlighted the need for infrastructure development to better diagnose, treat and contain them. Lassa fever virus (LASV) circulates within the same regions as EBOV, but rather than being sporadic it results in seasonal HF outbreaks, and is endemic in Guinea, Liberia, Sierra-Leone and Nigeria [15,16]. Misdiagnosis and incorrect discharge or admittance to an HFV treatment ward has very clear negative implications. LASV diagnosis was confounded during the EBOV epidemic [17]. Diagnostic test development for LASV is complicated by the level of diversity between circulating strains and most laboratories use in-house assays [18,19,20]. In fact, HFV diagnosis for the above-mentioned virus families is often based on in-house protocols, of which there are many variants [21,22,23,24]. There can be no doubt that the availability of an international standard to harmonise data reporting between this broad range of assays, allowing determination and comparison of sensitivity and cut-offs, is required.

The clinical significance of HFVs and the need for treatments and vaccine development is recognised through the inclusion of LASV, EBOV, Marburg virus, Rift Valley fever virus (RVFV) and Crimean-Congo haemorrhagic fever virus (CCHFV) on the WHO R&D blueprint and the U.K. vaccine network lists of priority pathogens [25,26]. Only HFVs caused by members of the *Flaviviridae* family, Dengue virus and Yellow fever virus have licensed vaccines available for use globally [27,28]. For those HFVs included on priority pathogen lists, there are several candidate vaccines currently under development, but only those targeting filoviruses have reached clinical trials [29,30,31]. Filovirus vaccines were subject to accelerated development, with close ethical and regulatory review overseen by the WHO in partnership with The African Vaccine Regulatory Forum [32,33]. Two candidates progressed to phase III clinical trials and were awarded emergency use licences, however only one study reached completion [34]. This is a departure from the traditional research and development pathway, which typically takes many years, and the framework can now be applied to other HFVs. It is envisaged that candidate vaccines being developed against LASV [16,19], CCHFV [23,35] and RVFV [36] may be taken through early human clinical trials, demonstrating safety, with stockpiles then made available for emergency use while sufficient clinical trial data is collected for licensure. LASV vaccine availability has been made a priority, and funding awarded to developers by the Coalition for Epidemic Preparedness Innovations (CEPI) [37], with other HFV identified for future development funding [38]. LASV is estimated to be contracted by 100,000–300,000 people every year across endemic countries in Western Africa and accounts for approximately 5000 deaths, although the case fatality rate can reach 15% in hospitalised patients [39]. It is thought the population at risk of LASV amounts to 58 million, with recorded cases in Nigeria during 2018 and 2019 greatly exceeding those from previous years [40,41,42]. Within a vaccine target product profile (TPP) published by the WHO, it is highlighted that there is a need for sero-surveillance to provide updated and more accurate estimates on the incidence, seroprevalence and geographic distribution of LASV [43]. Serological assays will play an important role in the measurement of clinical and pre-clinical immunogenicity during vaccine studies. This is also the case for CCHFV and RVFV vaccine development. While CCHFV has a lower incidence than LASV, infecting 10,000–15,000 people each year, it is endemic across a far larger geographical area, having been reported in Africa, the Balkans, the Middle East and Asia, with the vector present in countries south of the 50° parallel north [35,44]. Thus, 3 billion people are predicted to be at risk of CCHFV infection. The WHO is currently working to formulate a TPP for CCHFV vaccine development [45]. Outbreaks of RVFV have so far only been documented in sub-Saharan Africa and Egypt, with 1 billion people at risk of infection, presenting a significant economic burden due to loss of livestock [46,47]. In all cases, the correlate of protection for these HFVs is not defined, thus it is important to be able to compare data reported across different serological assay platforms. This is achieved by correlating results to a common reference reagent for antibody to provide the same readouts and normalise data from assays. The availability of a WHO International Standard early on in the vaccine development process is, therefore, considered beneficial.

## 3. Current Status of WHO Standards for Haemorrhagic Fever Viruses

Reference materials are required to support the development of reliable diagnostic kits, to evaluate vaccine and treatment efficacy and monitor immune responses to HFVs. As mentioned, there are different types of standards that are used to offer assurance at different stages in the detection of an infection (Table 1). Here, we will focus on the WHO reference materials to be used for the calibration of assays; their availability and use allows comparability of assays worldwide through harmonisation of the data by expressing the result to the same unitage (International unit) [4].

### 3.1. Diagnostics

Speed and accuracy of diagnosis of the aetiological agent responsible for the HF is essential. Molecular techniques are the preferred choice by diagnostic laboratories; these are usually based on the amplification of the nucleic acid sequences of the viruses of interest and allow for great specificity and sensitivity. All of the components required for the molecular detection of HFVs are suitable for inclusion in mobile and on-site point-of-care laboratories. These diagnostic settings have been proven to be very important in outbreak scenarios in Africa, where a timely response to HF cases is difficult to achieve solely by the local public health infrastructures [10,12,13]. A reference material for nucleic acid amplification technique (NAAT)-based assays will contain the target viral sequences of the specific microorganism; the ideal standard is the whole virus, made non-infectious by a validated inactivation procedure. An example is the 1st WHO International Reference Panel for Dengue virus types 1 to 4 RNA [48]. Chosen laboratory strains for each of the four Dengue types were grown in insect cells, heat-inactivated and 2000 vials for each candidate prepared as lyophilised material. The use of this international standard reduced inter-laboratory variation and harmonised data in a multi-centre, international collaborative study to evaluate the candidate preparations. This is the only WHO International Standard established for HFVs (Table 2). The major obstacle in generating reference preparations for HFVs is that most of them are classified as hazard/risk group 4 [49,50,51,52], restricting the handling of these viruses to those laboratories with biological safety level 4 (BSL4) facilities. For EBOV, a further issue was proving inactivation using methods that are compatible with downstream detection by nucleic acid amplification. Although heat inactivation of EBOV clinical samples with a moderate viral load has been shown to completely inactivate the virus [53], analysis conducted during the Western Africa outbreak in 2013–16 revealed that a two-step inactivation is required to make high titre EBOV material safe [54]. As an alternative, in 2015 the WHO Expert Committee on Biological Standardisation (ECBS) established reference reagents for Ebola RNA assays based on Ebola RNA sequences packaged into human immunodeficiency virus (HIV)-like particles [55]. These particles are safe, as they do not contain any functional HIV or EBOV gene sequences; the particles are also not infectious as they lack an envelope protein on their surface [56]. This material was evaluated in a collaborative study involving 13 laboratories worldwide and it was shown to harmonise results obtained by different methods (Figure 1, [57]). The chimeric HIV-EBOV platform offers a control for all the steps of the sample analysis, from extraction to the final amplification step; furthermore, it is flexible and can be adapted to other RNA viruses as soon as the viral sequence is known, and can thus be produced in a short amount of time (1–2 months). This represents an ideal tool for preparedness in the event of an outbreak caused by an unknown virus—Disease X [26]. However, no clinical samples were included in this study and more investigation is needed to assess how well these preparations compared to the real virus.

### 3.2. Serological Standard

A reference material for determining an antibody response has many applications in public health. Serology can be used as a diagnostic tool, *per se*, or in combination with NAAT assays or an antigen-based test. Serological tests are used to confirm previous exposure to the pathogen, in epidemiology studies, surveillance programmes, to check the immune response to a vaccine and to determine potency of antibody treatments, either as cocktail of purified antibodies or passive transfer of convalescent plasma. The 1st WHO International Standard for an antibody against an HFV is the yellow fever virus (YFV) anti-serum, developed in 1962 by the Statens Seruminstitut [60,63]. The preparation is a pool of serum collected from three monkeys immunised with the Asibi strain of YFV, which was evaluated in an international collaborative study involving 11 laboratories located in 10 countries [63]. This preparation has been used to compare the immune response in animal models for the testing of YF vaccines, and for potency assays for anti-YF sera. The use of animal sera to develop a reference reagent has the drawback of potentially reduced commutability, i.e., resemblance with the clinical sample [64]; lack of commutability of a reference preparation may result in a lack of harmonisation or even increase the inter-laboratory variance [65]. Moving forward for antibody standards, the use of human convalescent plasma and serum is the preferred choice; however, for HFVs this has its own issues, mainly the difficulty in sourcing this type of material from countries where an epidemic is taking place with political, anthropological and safety concerns [66]. During the 2013–16 Ebola outbreak, a panel of convalescent plasma or serum was produced from samples donated by repatriated health workers from Norway, the United States and the United Kingdom who had contracted EBOV in Western Africa. This was evaluated in an international collaborative study for the establishment of an interim standard to help harmonise serological assays for EBOV until a full WHO IS was available. Convalescent plasma donated by the America Red Cross was established by the WHO ECBS in 2015 as a reference reagent for Ebola virus antibody, due to its ability to reduce the inter-laboratory variance when the results were normalised to it [55]. A follow-up analysis of the data collected from the collaborative study highlighted how performance of the different methods varied greatly, and that there was a need for an international standard to align assays [67]. Two years later, the WHO International Standard for Ebola virus antibody and an international reference panel were established by the WHO ECBS [58]. The IS is a pool of convalescent plasma from six Sierra Leone patients recovered from Ebola infection and has an assigned potency of 1.5 IU/mL. The preparation was evaluated against the interim preparation, a panel of convalescent serum and plasma and two monoclonal antibodies by 17 laboratories from 4 countries. The IS had the highest absolute titre among the samples evaluated and the use of the IS increased harmonisation of the results from laboratories by reducing the coefficient of variation between laboratory datasets (Figure 2). Similar preparations for other HFV have been endorsed by ECBS and are in development (Table 2).

### 3.3. Antigen Standard

NAAT and antibody detection methods are the preferred systems used by clinical laboratories. These require good laboratory infrastructure, such as specialised equipment, a constant electricity supply and trained personnel, which may not be available in the field. The use of rapid point-of-care tests, which are usually based on antigen detection, are extremely important to support a quick response, especially in rural areas [9,69]. In the case of EBOV, there are a few products that have been authorised for emergency use [18]. For some HFVs there are antigen-based assays under development, but these are still designated for research use only [19,70,71]. Yet for other significant pathogens, such as CCHF, no commercial kits are available [23]. This lack of assays is also reflected in the paucity of IS for antigen detection. Despite the majority of these assays being based on a positive/negative response, for which the need of a calibrator may not appear obvious, a reference preparation is still required to determine the sensitivity of these assays. This ensures that the limit of detection for different assays is comparable. While there is a reference preparation available in the WHO catalogue for EBOV (Table 2), this is not currently available for other HFVs. The reference preparation for EBOV is a panel of recombinant Ebola VP40 proteins produced in *E. coli* and diluted in human serum at different concentrations [55]. The panel allows for qualitative assessment of the performance of point-of-care rapid tests for EBOV. It is recommended that antigen standards will be developed for other HFVs in light of the increased importance of point-of-care tests.

### 3.4. Vaccine Standard

Many vaccines are under development for HFVs, but only two have been licensed so far. For YFV, there are 6 manufacturers producing different vaccines that are all based on the live attenuated YFV strain 17D [28]. The WHO recommends that a vaccine’s potency is calibrated to the first international standard for yellow fever vaccine and expressed in International Units [72,73]. Furthermore, the use of the International Standard is recommended for the certification of new batches of YF vaccines by the national control laboratories, to guarantee they meet the potency required by the regulatory body. It is expected that similar standards will be available for the other HFVs once a vaccine has been licensed. The current Dengue vaccine, Dengvaxia from Sanofi Pasteur, has been licensed in 20 countries worldwide [27], but there are still concerns regarding its safety [74]. Two EBOV vaccines have been licensed nationally in China under the animal rule [75], and in Russia based on Phase I/II clinical trial data [76]. A further two vaccines are under review by WHO for emergency use authorisation listings and by the FDA; they are undergoing or have completed phase III clinical trials [29]. For LASV there are currently no licensed vaccines, but WHO has produced a target product profile for vaccines targeting this virus [43]. A similar document is expected to be finalised for other viruses on the WHO R&D Blueprint list of priority pathogens, such as CCHF and RVFV. While there is a vaccine for RVFV for veterinary use, one has not been licensed for human use [47].

## 4. Future Outlook

As we move into new areas of research and development for HFVs, there is an increasing need to develop robust assays to measure the efficacy of a variety of treatments and therapies [77]. Recent initiatives, such as the WHO R&D Blueprint and the subsequent funding available through bodies such as CEPI and Innovate UK [78], have emphasised and supported the need to focus on priority pathogens for vaccine development. Reliable diagnostics also need to be in place for the evaluation of treatments and to quickly identify and contain HFV outbreaks. Organisations such as the Foundation for Innovative New Diagnostics (FIND) play a central role in supporting the development and implementation of new diagnostic tools [79]. Early implementation of reference standards and the consequent adoption of the International Unit to calibrate assays has so far not been realised before reaching the point of a PHEIC announcement. In these cases, there is simply not enough time to produce the materials contemporaneously to the outbreak. The main issues are sourcing of sera and plasma to make an antibody standard from affected countries [66] or even to isolate the virus and make its sequence public in the case of standards for nucleic acid testing. An international standard can take 2 to 3 years to be established [80]. Therefore, these preparedness initiatives are important in order that vaccines and their associated measurement assays (and standards) can be developed through to a phase I/II clinical trial ahead of an outbreak. The ability to compare vaccine candidates for HFVs through assay harmonisation with a reference reagent will facilitate the identification and selection of the most promising vaccines to take forward into large and expensive clinical trials (phase III), and therefore may provide economic and resource benefits. Moreover, the use of standards in assays can help smooth the path through the regulatory process. Indeed, WHO guidelines advise that if a standard is available, preferably the International Standard, then it should be used for assay calibration and reporting of results [81]. It is hoped that these approaches can increase the understanding and knowledge of the benefits of standards to the end user and that they are more widely adopted, leading to better diagnosis and improved vaccines and therapies. 

## Figures and Tables

**Figure 1 viruses-11-00781-f001:**
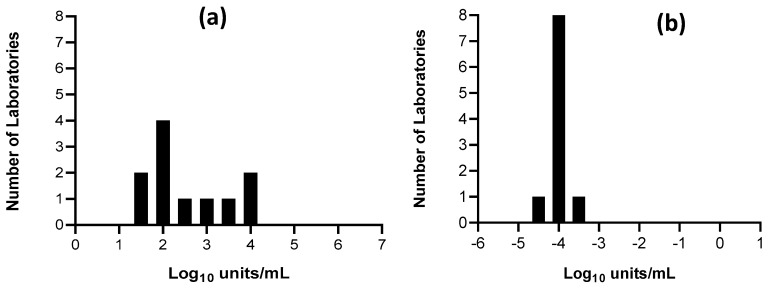
Harmonisation of the potencies for Ebola RNA samples after assay calibration to WHO reference reagent. Mean laboratory estimates for an EBOV RNA incorporated into an HIV-like particle using NAAT methods targeting EBOV np, gp, or vp35 genes. (**a**) Results were reported by the participants as Log10 “detectable units”/mL, being “copies” for majority of the quantitative assays, or detection limit by Ct values. (**b**) Mean estimates of the same samples expressed as relative to the WHO reference reagent for EBOV RNA. Values are reported as Log10 WHO units/mL. Full details of the study are available in the WHO report [57].

**Figure 2 viruses-11-00781-f002:**
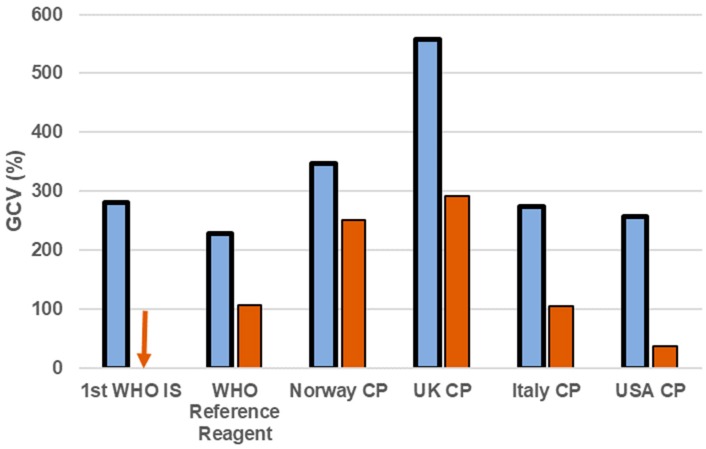
Increased harmonisation of EBOV neutralizing antibody titres by the WHO International Standard. Geometric coefficient of variation (GCV) was calculated using the neutralizing antibody titres against EBOV as reported by the participants (50% neutralisation titre, blue bars) or after normalisation to the International Standard (1st WHO IS), and potency is expressed as International Unit (IU, orange bars). Note: CP = convalescent plasma. Arrow down = GCV:0% where the value has been assigned. Full details are available in the final report of the collaborative study [68].

**Table 1 viruses-11-00781-t001:** Type of standards and their applications.

	WHO International Standard	External Controls (Secondary Standard)	Internal Standard/In-Run Control
Sample Quantification	yes	yes	yes
Assay Performance (over time/between operators)	yes	yes	no
Data Comparison (between labs/assay)	yes	yes/no	no
Assay Calibration in International Unit	yes	no	no

**Table 2 viruses-11-00781-t002:** WHO reference preparations for haemorrhagic fever viruses available in the WHO catalogue.

Pathogen	RNA	Antibody	Antigen	Vaccine	Reference
Ebola virus	2015	2017	2015	-	[55,58]
Sudan virus	Endorsed 2018	Endorsed 2018	-	-	[59]
Marburg virus	endorsed 2018	endorsed 2018	-	-	[59]
Yellow fever virus	-	1962	-	2003	[60,61]
Dengue virus	2016	endorsed 2017	-	-	[58,62]
Lassa virus	Endorsed 2018	Endorsed 2018	-	-	[59]
CCHF virus	endorsed 2018	endorsed 2018	-	-	[59]
RVF virus	proposed 2019	proposed 2019	-	-	

Note: CCHF = Crimean-Congo haemorrhagic fever; RVF = Rift Valley fever; - = not available.

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
