# Peer review of "The Role of Reference Materials in the Research and Development of Diagnostic Tools and Treatments for Haemorrhagic Fever Viruses"

_viruses, 2019, doi:10.3390/v11090781_

Round 1
Reviewer 1 Report
In the review by Mattiuzzo et al., the authors describes the role of reference materials and various standards in diagnostics and treatment of emerging haemorrhagic fever causing viruses (HFV). The review is quite timely as the current Ebola outbreak raging in the Democratic Republic of the Congo, and other HFV outbreaks occuring elsewhere. The review also quite comprehensive. I find no major weakness in this report as currently constructed.
Author Response
We would like to thank the reviewer for reading our review and for the positive comments.
Reviewer 2 Report
Manuscript ID: viruses-560773
Type of manuscript: Review
Title: Role of reference materials in the research and development of treatments for Haemorrhagic Fever Viruses
Authors: Giada Mattiuzzo, Emma M Bentley, Mark Page
This manuscript seeks to overview recent development in the design of standards for diagnostics, therapeutics, and vaccines against highly pathogenic RNA viruses. After 2013-2016 EBOV outbreak in West Africa, this field received substantial support from different national and international organizations. Since 1890s, it is well known that biological reference materials are crucial for development of diagnostics and biologics. There are several challenges for development of reference materials for highly pathogenic RNA viruses since handling of these infectious agents requires high biological containment facilities (BSL4/ABSL4). These challenges have been recently summarized and published in Feb issue of Emerging Infect Dis by Rampling T, Page M, Horby P under title “International Biological Reference Preparations for Epidemic Infectious Diseases”. In published review and in the submitted manuscript, the authors are focusing on the WHO Blueprint priority pathogens (EBOV/MARV, LASV, Nipah/MERS, CCHF, and etc.) and using EBOV as example of some progress in development of reference biological materials. With development of HIV-like particles carrying some EBOV RNA sequences, the WHO established the 1st EBOV reference reagent for nucleic acid (NA) based diagnostic assays, e.g, RT/PCR.
Major points:
1. Line 67-69: “Reporting biological activity in physical/chemical units such as mg or copies therefore is neither relevant nor appropriate”.
Meanwhile, near-patient NA-based diagnostic assays are becoming the most widely used diagnostic tools for EBOV and LASV infections. The WHO EBOV RNA VP40-L reference reagent has an assigned unitage of 7.7 log10 units per vial (~50,000,000 units/vial). It was designed by subcloning limited EBOV genes and packaging them in lentivirus-like particles. “Units” are quantitative copies of EBOV RNA sequences.
2. While NA-based assays seem to be the most promising near-patient diagnostic approach, reference materials based on RNA copies have no relation to biological (infectious) activity. In fact, EBOV and LASV are negative-strand viruses, and their RNA molecules are not infectious, in contrast to positive-strand RNA viruses (e.g., flavi, and alphviruses). In addition, quantitative analysis revealed a huge variation among hemorrhagic viruses in RNA/PFU and particle/RNA ratios (Weidman et al., 2011; Alfson et al., 2015), even for the same viral species (e.g., EBOV). Discussion of all these issues crucial for development of reference standards are completely missing in the submitted review.
Minor point:
In the manuscript, the authors mentioned in different places development of biological references for vaccine development. Meanwhile, the title is focusing only on “treatment” of infectious caused by HFVs.
Author Response
Major points:
Line 67-69: “Reporting biological activity in physical/chemical units such as mg or copies therefore is neither relevant nor appropriate”.Meanwhile, near-patient NA-based diagnostic assays are becoming the most widely used diagnostic tools for EBOV and LASV infections. The WHO EBOV RNA VP40-L reference reagent has an assigned unitage of 7.7 log10 units per vial (~50,000,000 units/vial). It was designed by subcloning limited EBOV genes and packaging them in lentivirus-like particles. “Units” are quantitative copies of EBOV RNA sequences.
Authors’ reply: We would like to emphasise that there is no conversion factor between Unit and copies. To generate the WHO EBOV RNA reference reagents an in-house characterization was carried out to guarantee that the material could have acted as calibrator and therefore allowed for 5 to 6 serial dilution steps. This was reported in Mattiuzzo et al 2015 as copies/mL. However, the value of Unit/mL assigned by the WHO ECBS following the International Collaborative Study was different from the copies number calculated. The unitage is usually assigned arbitrarily, however, an attempt is made to have the unit/mL in the same order of magnitude of the results provided by the 27 methods used because this facilitates reporting of the data in integers that are likely to be recorded by the assay.
We understand how having a unit/mL value similar to the “copies” or other physical measurement may lead people to assume equivalence between the two and therefore search for a conversion factor. Therefore, we have added the following sentence in line 69 to strengthen the concept:
In some cases, the assigned unitage of a reference preparation in unit/mL is arbitrarily chosen to be similar to the value of the physical units; the reason for this is to facilitate the end user in the transition from the metrology system in use (mg/mL, copies/mL, etc.) to the potency expressed in unit/mL.
While NA-based assays seem to be the most promising near-patient diagnostic approach, reference materials based on RNA copies have no relation to biological (infectious) activity. In fact, EBOV and LASV are negative-strand viruses, and their RNA molecules are not infectious, in contrast to positive-strand RNA viruses (e.g., flavi, and alphviruses). In addition, quantitative analysis revealed a huge variation among hemorrhagic viruses in RNA/PFU and particle/RNA ratios (Weidman et al., 2011; Alfson et al., 2015), even for the same viral species (e.g., EBOV). Discussion of all these issues crucial for development of reference standards are completely missing in the submitted review
Authors’ reply: We agree with the reviewer that diagnostic assays, based on the detection of either antigen or viral nucleic acid, target circulating virus particles without distinguishing between infectious or defective particles. The two measurements are quite different. For our reference preparation, we have identified a ratio of 10:1 between physical particles and particles containing a viral genome (Mattiuzzo et al 2015). The reference preparation is not infectious, therefore the ratio to infectious particles is not possible to determine. Ultimately, this is an intrinsic issue with the assay. However, a reference preparation has to resemble the clinical sample and behave like one in the assays used. It will harmonise assays with different output results, but it will not solve issues associated with the methods. This is actually an important point and we would like to thank the reviewer for bringing this to our attention. We have added the following sentence to clarify the limitations of a reference preparation in line 76:
Furthermore, a limitation on the use of a reference material is that it will not solve substantial issues related to the assay. For instance, if the sensitivity of an assay is very low, the use of a reference material may highlight this potential problem, but it will not correct a false negative result.
Minor point:
In the manuscript, the authors mentioned in different places development of biological references for vaccine development. Meanwhile, the title is focusing only on “treatment” of infectious caused by HFVs.
Authors’ reply: we appreciate the reviewer’s comments. Vaccine development is one of the potential treatments, and the one which will benefit the most in having an antibody reference material. There are other treatments which will benefit of calibrated serological assays. To include this, we have now added:
Line 209-210: vaccine and to determine potency of antibody treatments, either as cocktail of purified antibodies or of convalescent plasma (REFERENCE: PMID: 28355507)
Furthermore, in the review we have also mentioned the impact on reference material on development of diagnostics, and therefore we have changed the title to:
“Role of reference materials in the research and development of diagnostic tools and treatments for Haemorrhagic Fever Viruses”
We have also corrected the main text as follows:
line 111: treatments and vaccines development
Round 2
Reviewer 2 Report
In the revised manuscript the authors properly addressed the reviewer' critiques and comments.